# Guiding Diffusion Models with Reinforcement Learning for Stable Molecule Generation

## Abstract

Generating physically realistic 3D molecular structures remains a core challenge in molecular generative modeling. While diffusion models equipped with equivariant neural networks have made progress in capturing molecular geometries, they often struggle to produce equilibrium structures that adhere to physical principles such as force field consistency. To bridge this gap, we propose **Reinforcement Learning with Physical Feedback (RLPF)**, a novel framework that extends Denoising Diffusion Policy Optimization to 3D molecular generation. RLPF formulates the task as a Markov decision process and applies proximal policy optimization to fine-tune equivariant diffusion models. Crucially, RLPF introduces reward functions derived from force-field evaluations, providing direct physical feedback to guide the generation toward energetically stable and physically meaningful structures. Experiments on the QM9 and GEOM-drug datasets demonstrate that RLPF significantly improves molecular stability compared to existing methods. These results highlight the value of incorporating physics-based feedback into generative modeling.

## 1 Introduction

Recent advancements in generative models have demonstrated remarkable potential for generating diverse and high-quality molecular structures. Among these, diffusion models (Ho et al., 2020) have emerged as a prominent area of research in molecular generation due to their superior generative capabilities and theoretical soundness. While other generative models, such as Generative Adversarial Networks (GANs) (Goodfellow et al., 2020) and Variational Autoencoders (VAEs) (Kingma et al., 2019), have also made significant progress, diffusion models have shown particularly compelling performance in generating complex molecular structures.

Integrating these models with equivariant graph neural networks (Satorras et al., 2021; Liao & Smidt, 2022; Thomas et al., 2018) further enhances their performance by explicitly considering the geometric properties and physical constraints of molecules (Xu et al., 2022; Jing et al., 2022). This combination allows for improved generation of molecules with desired properties, as equivariant graph neural networks ensure equivariance to rotations, translations, and reflections, resulting in physically more realistic and stable conformations. Building upon these advancements, Equivariant Diffusion Models (EDMs) (Hoogeboom et al., 2022) have emerged as particularly promising within this landscape. A key advantage of EDMs lies in their ability to operate on both continuous (3D conformation) and categorical features (atom types), rather than solely focusing on generating molecular conformations. This enhanced capability makes EDMs particularly well-suited for de novo drug discovery, where precise control over molecular properties and functionalities is essential.

Despite the successes of the aforementioned approaches, we observe a notable limitation: the stability of the generated molecular structures. Specifically, when evaluating generated conformations using physical force fields, we frequently observe high residual atomic forces (as illustrated in Figure 1), indicating significant strain and instability. This suggests that while the models may produce chemically valid molecules, they often fail to generate physically plausible and energetically favorable conformations.

This naturally raises a crucial question: *how can we guide generative models towards producing more stable molecular structures?* Inspired by recent advances in Large Language Models (LLMs), particularly Reinforcement Learning from Human Feedback (RLHF) (Stiennon et al., 2020), we

explore a novel paradigm for training molecular diffusion models. Traditional diffusion models are trained via maximum likelihood estimation, akin to Supervised Fine-Tuning (SFT) in LLMs. However, RLHF shows that reward-based fine-tuning can dramatically improve alignment with human or domain-specific preferences.

Drawing this analogy, we propose a new approach: **Reinforcement Learning with Physical Feedback (RLPF)**, which integrates reinforcement learning with equivariant diffusion models using physically grounded rewards. Specifically, RLPF leverages reward signals derived from force field-based metrics to fine-tune pretrained diffusion models, thereby encouraging the generation of physically realistic and energetically stable molecules. These signals can be computed from classical force fields, quantum mechanical approximations, or other structure-informed heuristics, and serve as a domain-specific counterpart to human feedback in RLHF.

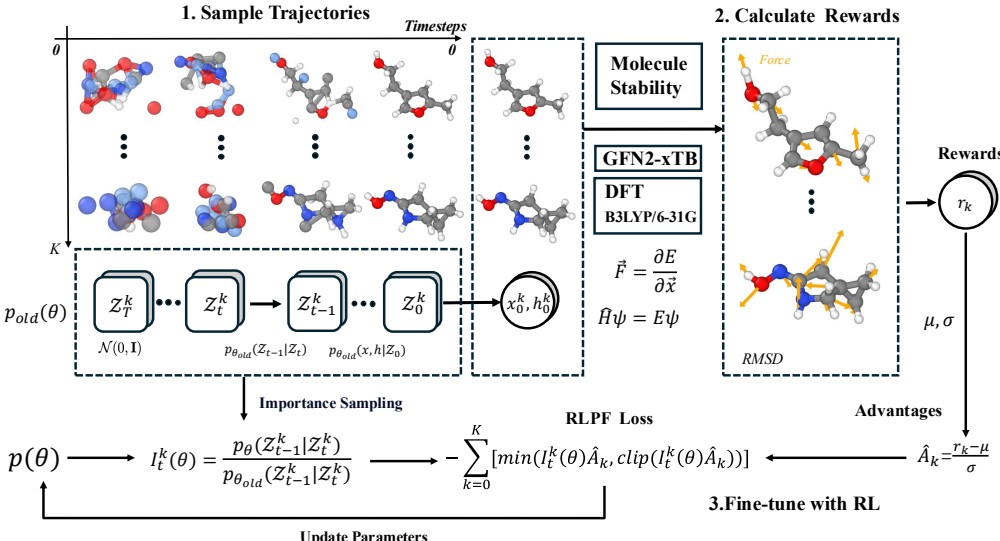

Figure 1: Overall workflow of the RLPF algorithm. RLPF fine-tunes a pretrained diffusion model for molecular generation in three steps. First, the model $p_\theta$ generates molecular trajectories. Second, molecule stability is evaluated using force field metrics such as classical or quantum energy gradients. Unstable molecules typically exhibit large residual forces. Third, reinforcement learning is used to refine the model using a PPO-style policy gradient, guided by computed rewards and advantage estimates.

RLPF formulates the denoising process in diffusion models as a Markov Decision Process (MDP) (Puterman, 1990). Each reverse step of the diffusion process corresponds to an action within the MDP, and the reward is assigned at the final denoising step based on the physical plausibility of the generated molecule. As shown in Figure 1, upon reaching the terminal state, the molecule is evaluated for physical stability (e.g., force residual), and this reward is used to optimize the model parameters via the REINFORCE algorithm (Williams, 1992), with further stabilization via PPO-style clipping.

RLPF is model-agnostic and can be applied to any diffusion-based molecular generation framework. In this work, we instantiate RLPF on Equivariant Diffusion Model (EDM (Hoogeboom et al., 2022)), which have demonstrated state-of-the-art performance. We further validate the generalizability of RLPF by applying it to GeoLDM (Xu et al., 2023) and UniGEM (Feng et al., 2024). We evaluate our method on two benchmark datasets: QM9 (Ramakrishnan et al., 2014) for small organic molecules, and GEOM-drug (Axelrod & Gomez-Bombarelli, 2022) for drug-like molecular structures. Empirical results show that RLPF significantly improves the quality of the generated molecules, outperforming both baseline EDM and supervised fine-tuned variants.

**Contributions:**

1. We propose **RLPF**, a novel method that integrates physics-informed reinforcement learning with equivariant diffusion models, using force field-based feedback for 3D molecular generation.

2. RLPF substantially improves the quality of the molecules generated in both the QM9 and GEOM drug, achieving better performance than existing diffusion-based methods in multiple stability and validity metrics.

3. RLPF is model-agnostic and demonstrates strong generalizability, effectively improving generation quality when applied to diverse backbones.

## 2 RELATED WORK

In the field of 3D molecular generation, generative models can be broadly classified into two categories: autoregressive models and diffusion models.

**Autoregressive models** generate molecules sequentially, atom by atom or bond by bond, where each generation step is conditioned on the previously generated substructure. This sequential nature allows for precise control over the molecular construction process. Early works in this area often employed recurrent neural networks (RNNs) to generate SMILES strings, a linear representation of molecular structures (Gómez-Bombarelli et al., 2018). However, SMILES-based approaches can suffer from issues related to canonicalization and the difficulty of capturing 3D structural information. More recent approaches have focused on generating molecular graphs directly, using geometrical graph neural networks to represent and process molecular structures. These graph-based autoregressive models generate molecules by iteratively adding nodes (atoms) and edges (bonds) to the growing graph (Gebauer et al., 2019; Daigavane et al., 2023).

**Diffusion models** offer an alternative approach to molecular generation by learning to reverse a noise corruption process. These models operate by progressively adding noise to a data distribution (e.g., molecular structures) until it becomes a simple, tractable distribution (e.g., Gaussian noise). The model is then trained to learn the reverse process, i.e., denoising, allowing it to generate new samples by iteratively removing noise from the simple distribution. In the context of molecular generation, diffusion models have been applied to various molecular representations, including point clouds, graphs, and voxel grids. Xu et al. (2022) introduced GeoDiff, a diffusion model specifically designed for 3D molecular conformation generation in the Euclidean space. TorsionDiff (Jing et al., 2022) applies the diffusion process over the torsion angles and leaves the other degrees of freedom fixed. Hoogeboom et al. (2022) proposed EDM, where the model learns to denoise a diffusion process that operates on both continuous coordinates and categorical atom types. Equivariant neural diffusion (EDN) (Cornet et al., 2024)generalizes EDM, by defining the forward process through a learnable transformation and extending the flexibility of the hidden state in the diffusion model. Igashov et al. (2024) investigated the use of equivariant neural networks within the diffusion framework for molecular linker design. Equivariance ensures that the model's predictions are consistent with the underlying symmetries of the molecular system (e.g., rotations and translations), leading to more physically plausible generated structures. Diffusion models, while computationally more demanding than autoregressive models, have demonstrated the ability to generate high-quality and diverse molecular structures.

While generative models for molecules have demonstrated the ability to produce reasonably stable structures, they still lag behind the advancements of deep learning in natural language processing (NLP) and computer vision (CV), as highlighted in (Zhao et al., 2023). Notably, **reinforcement learning** (RL) has proven effective in fine-tuning diffusion models for text-to-image generation (Black et al., 2023; Fan et al., 2024), enabling these models to leverage feedback and improve performance. Similarly, RL has been successfully applied to fine-tune autoregressive molecular generation models (Hastrup & Bhowmik, 2024) and 2D graph diffusion models (Liu et al., 2024). However, a key gap remains: effective methods for fine-tuning molecular generative diffusion models using RL are still lacking. Consequently, these models have yet to fully exploit their own generated data as feedback to enhance molecule stability and ensure alignment with force-field-stable structures.

# 3 PRELIMINARIES

This section introduces the foundational concepts of equivariant diffusion models and outlines the reinforcement learning formulation adopted in the DDPO framework.

## 3.1 EQUIVARIANT DIFFUSION MODEL

The Equivariant Diffusion Model (EDM) (Hoogeboom et al., 2022) generates 3D molecules while respecting E(3) symmetries (translation, rotation, reflection) via an EGNN-based denoiser (Satorras et al., 2021). Given atom coordinates $\mathbf{x}$ and features $\mathbf{h}$, EDM learns a joint denoising process over $\mathbf{z}_t = [\mathbf{x}_t, \mathbf{h}_t]$ with Gaussian forward noise:

$$q(\mathbf{z}_t \mid \mathbf{x}, \mathbf{h}) = \mathcal{N}_{xh}\big(\mathbf{z}_t \mid \alpha_t[\mathbf{x}, \mathbf{h}], \sigma_t^2 \mathbf{I}\big), \tag{1}$$

and enforces equivariance

$$p(\mathbf{y} \mid \mathbf{x}) = p(R\mathbf{y} \mid R\mathbf{x}) \quad \text{for any orthogonal } R. \tag{2}$$

An EGNN $\phi$ predicts noise at step $t$ to $t-1$,

$$\hat{\epsilon}_t^{(x)}, \hat{\epsilon}_t^{(h)} = \phi\big(z_t^{(x)}, [z_t^{(h)}, t/T]\big) - [z_t^{(x)}, 0], \tag{3}$$

and is trained by a weighted denoising objective (SNR weighting omitted for brevity),

$$\mathcal{L} = \mathbb{E}_{\epsilon \sim \mathcal{N}(0, \mathbf{I})}\Big[\|\epsilon - \phi(\mathbf{z}_t, t)\|^2\Big]. \tag{4}$$

Sampling starts from standard Gaussian noise and iteratively applies the reverse transition $p(\mathbf{z}_{t-1} \mid \mathbf{z}_t)$:

$$\mathbf{z}_s = \frac{1}{\alpha_{t|s}}\mathbf{z}_t - \frac{\sigma_{t|s}^2}{\alpha_{t|s}\sigma_{t|s}}\phi(\mathbf{z}_t, t) + \sigma_{t \to s}\epsilon, \quad s = t - 1, \tag{5}$$

until $t = 0$, yielding final coordinates $\mathbf{x}$ and features $\mathbf{h}$ that define the molecule.

## 3.2 DENOISING DIFFUSION POLICY OPTIMIZATION

DDPO (Black et al., 2023) formulates the diffusion sampling process as a multi-step Markov Decision Process (MDP), enabling policy gradient methods to optimize user-defined reward functions over generated samples.

The MDP is defined as:

- **State:** $s_t = (c, t, x_t)$, where $c$ is context, $t$ the timestep, and $x_t$ the latent at step $t$.
- **Action:** $a_t = x_{t-1}$, the output of the reverse diffusion step.
- **Policy:** $\pi(a_t|s_t) = p_\theta(x_{t-1}|x_t, c)$.
- **Reward:** $R(s_t, a_t) = r(x_0, c)$ if $t = 0$, and 0 otherwise.

A trajectory spans denoising steps from $t = T$ to 0, yielding final sample $x_0$. The training objective is to maximize the expected reward:

$$\nabla_\theta J_{\text{DDPO}} = \mathbb{E}\left[\sum_{t=0}^{T} \nabla_\theta \log p_\theta(x_{t-1}|x_t, c) \cdot r(x_0, c)\right]$$

To enable multiple updates per trajectory, an importance sampling estimator is introduced:

$$\nabla_\theta J_{\text{DDPO}} = \mathbb{E}\left[\sum_{t=0}^{T} \frac{p_\theta(x_{t-1}|x_t, c)}{p_{\theta_{\text{old}}}(x_{t-1}|x_t, c)} \nabla_\theta \log p_\theta(x_{t-1}|x_t, c) \cdot r(x_0, c)\right]$$

For stability, DDPO further adopts a clipped surrogate objective in the style of PPO, constraining policy updates across iterations.

---

**Algorithm 1** Reinforcement Learning with Physical Feedback (RLPF)

---

**Input:** pretrained diffusion model $p_{\theta_{\text{pre}}}$, diffusion model $p_\theta$, the old diffusion model $p_{\theta_{\text{old}}}$, the number of sampling trajectories $N$, the reward model $\mathcal{M}$, the time-steps $T$, the Advantage $A$, the importance sampling ratio $I_t^k$

Initialize $p_\theta = p_{\theta_{\text{old}}} = p_{\theta_{\text{pre}}}$

**while** $\theta$ not converged **do**

    Collect $N$ samples from diffusion model $p_\theta$:   $\mathcal{D} = \{(x, h, z_0, \cdots, z_t) \sim \pi_\theta(x, h|z_0)\pi_\theta(z_0|z_1)\cdots\pi_\theta(z_{T-1}|z_T)p(z_T)\}$

    Compute reward with reward model $\mathcal{M}: r = \mathcal{M}(x, h)$

    Compute the gradient $E_t\left[\sum_{k=0}^{K} \min\left(I_t^k(\theta)\hat{A}_t^k, \text{clip}(I_t^k(\theta), 1 - \epsilon, 1 + \epsilon)\hat{A}_t^k\right)\right]$ for each time-step $t$ and each trajectory $k$, update $\theta$

    $p_{\theta_{\text{old}}} = p_\theta$

**end while**

**Output: Fine-tuned diffusion model** $p_\theta$

---

# 4 RLPF: REINFORCEMENT LEARNING WITH PHYSICAL FEEDBACK

Although RLPF builds upon the general DDPO (Black et al., 2023) framework originally developed for vision tasks, its core contribution lies in the nontrivial adaptation of this paradigm to 3D molecular generation. Specifically, RLPF casts the denoising diffusion trajectory as a Markov Decision Process (MDP) over spatial molecular structures, and incorporates domain-specific reward functions based on physically grounded force-field evaluations, such as xTB or DFT. This adaptation is technically challenging due to the geometric equivariance, size variability, and chemical validity constraints unique to molecular systems—factors not present in typical visual domains. To the best of our knowledge, RLPF is the first approach to integrate reinforcement learning with diffusion models using physics-informed rewards for stable molecule generation.

## 4.1 PROBLEM STATEMENT

The pretrained diffusion model generates a sample distribution $p_\theta$ through a fixed sampling process for molecular generation. The goal of the equivariant denoising diffusion reinforcement learning framework is to optimize the reward function $r$, which is defined over the generated molecules.

The objective function for this optimization is defined as:

$$\mathcal{J}_{\text{RLPF}}(\theta) = \mathbb{E}_{(x,h)\sim p_\theta}[r(x, h)], \tag{6}$$

where atom coordinates $x$ and atom features $h$ are sampled from the final latent state $z_0$, i.e., the molecule generated by the diffusion process.

## 4.2 DENOISING AS A MARKOV DECISION PROCESS

To optimize $\mathcal{J}_{\text{RLPF}}$ using RL, the denoising process is formulated as a sequence of multi-step MDPs. The elements of this MDP are defined as follows:

$$s_t \triangleq (z_t, t),$$
$$\pi(a_t|s_t) \triangleq p_\theta(z_{t-1}|z_t),$$
$$R(s_t, a_t) \triangleq \begin{cases} r(x, h), & \text{if } t = 0, \\ 0, & \text{otherwise,} \end{cases} \tag{7}$$
$$P(s_{t+1}|s_t, a_t) \triangleq (\delta_{t-1}, \delta_{z_{t-1}}).$$

The sequence consists of $\mathcal{T}$ time steps, after which the process transitions to a termination state. The cumulative reward of each trajectory is equal to $r(x, h)$. Therefore, maximizing $\mathcal{J}_{\text{RLPF}}$ is equivalent to maximizing $\mathcal{J}_{\text{RL}}$ in this MDP.

### 4.3 POLICY GRADIENT ESTIMATION

The RLPF framework aims to optimize the expected reward over the denoising trajectories:

$$\mathcal{J}_{\text{RLPF}}(\theta) = \mathbb{E}_{(x,h)\sim p_\theta}[r(x,h)], \tag{8}$$

However, directly optimizing this objective is challenging due to the non-differentiability of reward functions and the sequential nature of the denoising steps. Instead, we adopt a policy optimization approach and follow the DDPO (Black et al., 2023) formulation, which views the denoising trajectory as a latent Markov Decision Process and applies Proximal Policy Optimization (PPO) for stable fine-tuning.

In particular, we use a PPO-style **clipped surrogate objective** , denoted as $\mathcal{L}_{\text{RLPF}}^{\text{CLIP}}(\theta)$, to guide the optimization. For each time step $t$ and trajectory $k$, we define the importance sampling ratio:

$$I_t^k(\theta) := \frac{p_\theta(z_{t-1}^k|z_t^k)}{p_{\theta_{\text{old}}}(z_{t-1}^k|z_t^k)}, \tag{9}$$

where $p_{\theta_{\text{old}}}$ denotes the diffusion model before the current update. The advantage estimate $\hat{A}_t^k$ is computed via standardization of the scalar reward:

$$\hat{A}_t^k := \frac{r^k(x^k, h^k) - \mu}{\delta}, \tag{10}$$

where $\mu$ and $\delta$ are the running mean and standard deviation of recent rewards across trajectories.

The PPO-style clipped surrogate objective is defined as:

$$\mathcal{L}_{\text{RLPF}}^{\text{CLIP}}(\theta) := \mathbb{E}_t \left[ \sum_{k=0}^{K} \min \left( I_t^k(\theta)\hat{A}_t^k, \ \text{clip}(I_t^k(\theta), 1 - \epsilon, 1 + \epsilon)\hat{A}_t^k \right) \right], \tag{11}$$

which ensures stable updates by penalizing large deviations from the current policy.

Although $\mathcal{L}_{\text{RLPF}}^{\text{CLIP}}(\theta)$ is not the true gradient of $\mathcal{J}_{\text{RLPF}}(\theta)$, it serves as a stable and effective proxy objective for gradient-based optimization. This formulation allows RLPF to improve molecular generation performance while avoiding issues such as reward overfitting and policy collapse.

### 4.4 REWARD FUNCTION FOR MOLECULAR GENERATION

To guide the diffusion model toward physically meaningful outputs, we use a reward function based on molecular force deviation. This metric evaluates how well the generated molecular conformations align with equilibrium configurations under a given force field.

We compute the Root Mean Square Deviation (RMSD) of atomic forces using two methods: quantum mechanical calculations at the B3LYP/6-31G(2df,p) level of theory and the semi-empirical GFN2-xTB force field (Bannwarth et al., 2019). The former offers high accuracy but is computationally expensive, while the latter enables efficient force estimation for large-scale generation. This reward reflects how close the generated molecule is to a physically relaxed structure. Formally, it is defined as:

$$r_{\text{force}} = \sqrt{\frac{\sum_{i=1}^{N}(f_{i_x}^2 + f_{i_y}^2 + f_{i_z}^2)}{3N}}, \tag{12}$$

where $f_{i_x}, f_{i_y}, f_{i_z}$ denote the $x$, $y$, and $z$ components of the predicted force on atom $i$, and $N$ is the total number of atoms in the molecule.

This physically grounded reward encourages the model to generate conformations that are not only chemically valid but also energetically favorable.

### 4.5 SIZE-INVARIANT LOG-LIKELIHOOD ESTIMATION

To accommodate variable-size molecular graphs and ensure consistent policy gradient estimation, we modify the computation of the reverse transition log-probability at each denoising step using a masking mechanism.

Specifically, under the assumption that the reverse transition $p(z_{t-1}|z_t)$ follows a Gaussian distribution, the log-probability $\log p(z_s \mid z_t)$ is defined as:

$$\log p(z_s \mid z_t) = -\frac{1}{2} \cdot \sum_i M_i \cdot d^{-1} \sum_j M_i \cdot \left( \frac{z_{i,j}^{(s)} - \mu_{ij}}{\sigma_{ij}} \right)^2 \tag{13}$$

where:

- $z_{i,j}^{(s)}$ denotes the $j$-th feature of the $i$-th atom in the denoised latent $z_s$,
- $\mu_{ij}$ and $\sigma_{ij}$ are the predicted mean and standard deviation from $z_t$,
- $d$ is the number of features per atom (e.g., 3D coordinates and atom-type encoding),
- $M_i \in \{0, 1\}$ is a binary mask indicating valid atoms in the molecule.

This masked average ensures that molecules with different numbers of atoms contribute equally and meaningfully to the policy objective, regardless of zero-padding or batch structure. Such size-invariant log-likelihood estimation is critical for stabilizing reinforcement learning in molecular settings, and is absent in prior DDPO implementations on vision or language tasks. We omit the Gaussian normalization constant $\log(2\pi\sigma^2)$, which cancels out when computing the importance-weighted policy ratio $I_t^k(\theta)$ (see Equation equation 9) during optimization.

## 5 EXPERIMENTS

In this section, we evaluate our proposed reinforcement learning framework RLPF on two standard molecular datasets: QM9 and GEOM-drug. We compare RLPF-enhanced models with a range of state-of-the-art generative baselines, including EDM (Hoogeboom et al., 2022), EDM-BRIDGE (Wu et al., 2022), GEOLDM (Xu et al., 2023), EDN (Cornet et al., 2024), GeoBFN (Song et al., 2024), and UniGEM (Feng et al., 2024). Our evaluations focus on key molecular quality metrics, such as atom stability, molecule stability, chemical validity, uniqueness, and novelty. We show that EDM-RLPF substantially improves generation performance across these dimensions. We also demonstrate that RLPF generalizes across model backbones, such as GeoLDM, by fine-tuning in latent space while preserving decoding fidelity. This highlights the flexibility of RLPF as a general reinforcement-based fine-tuning framework. Additional details—including training configurations, reward design, sampling strategies, ablation studies, and property-conditioned generation experiments—are provided in Appendix D.

Table 1: Evaluation metrics for 3D molecular generation on QM9: Atom stability (A), molecule stability (M), validity (V), and Validity×Uniqueness (V×U). **EDM-RLPF** fine-tunes the EDM model using DFT-calculated forces. Bold indicates best performance; underline indicates second-best.

| Model | A [%] ↑ | M [%] ↑ | V [%] ↑ | V×U[%] ↑ |
|---|---|---|---|---|
| EDM (Hoogeboom et al., 2022) | 98.70 | 82.00 | 91.90 | 90.7 |
| EDM-BRIDGE (Wu et al., 2022) | 98.80 | 84.60 | 92.00 | 90.7 |
| GEOLDM (Xu et al., 2023) | 98.90 | 89.40 | 93.80 | 92.7 |
| END (Cornet et al., 2024) | 98.90 | 89.10 | 94.80 | 92.6 |
| UniGEM (Feng et al., 2024) | 99.0 | 89.8 | 95.0 | **93.2** |
| GeoBFN (Song et al., 2024) | **99.08** | 90.87 | 95.31 | 92.96 |
| **EDM-RLPF (ours)** | **99.08** ± 0.05 | **93.37** ± 0.25 | **98.22** ± 0.15 | 92.87 ± 0.07 |
| Data (Ground Truth) | 99.00 | 95.20 | 97.70 | 97.70 |

### 5.1 MOLECULE GENERATION ON QM9

**Dataset** The QM9 dataset (Ramakrishnan et al., 2014) contains approximately 130k small organic molecules, with up to nine heavy atoms and up to 29 atoms including hydrogens. Following Anderson

et al. (2019), we divide the dataset into training, validation, and test sets with 100k, 13k, and 18k molecules, respectively.

**Experimental setup** Following the workflow outlined in Section F, we first trained an EDM model on the QM9 dataset to generate molecules with 3D coordinates and atom types. Our training configuration aligns with the original EDM paper, and full implementation details are provided in Appendix E.1.1. During the RLPF fine-tuning phase, sampling is conducted using the same denoising diffusion process as in the original EDM model, requiring no additional dataset beyond the pretraining data. The number of denoising time steps $T$ is set to 1000, with $K = 512$ sampled trajectories per epoch. We fine-tune the EDM model using RLPF with force deviation computed via DFT at the B3LYP/6-31G(2df,p) level. Notably, reward computation is performed entirely on CPU without GPU acceleration. To improve parallel efficiency, we adopt batch sampling and pipeline-parallel reward evaluation. Detailed hyperparameters and training setup are provided in Appendix E.2.

After fine-tuning, we evaluate molecular quality using four key metrics: atom stability (proportion of atoms with valid valency), molecule stability (proportion of fully stable molecules), validity (RDKit-filtered chemical validity), and Validity×Uniqueness. For each evaluation, we sample 10,000 molecules and report the mean over three independent runs.

**Results** As shown in Table 1, the EDM fine-tuned with RLPF achieves consistent improvements in all evaluation metrics. The molecular stability increases from 82.0% to 93.37%, and the atom stability reaches 99.08%, matching or exceeding prior state-of-the-art models. The validity increases to 98.22%, and the combined Validity×Uniqueness score of 92.87% suggests improved chemical quality while preserving the diversity of the sample. These results indicate that RLPF contributes to enhancing the quality of molecular generation. Additionally, EDM-RLPF achieves consistently high stability across different sampling steps; see Appendix D.3 for details.

Table 2: Evaluation metrics for 3D molecular generation on GEOM-drug. Atom stability and Validity. EDM-RLPF is fine-tuned using force deviation from GFN2-xTB. Bold indicates best performance, underline indicates second-best.

| Model | Atom Stability (%) ↑ | Validity (%) ↑ |
|---|---|---|
| EDM (Hoogeboom et al., 2022) | 81.3 | 91.9 |
| EDM-BRIDGE (Wu et al., 2022) | 82.4 | 91.9 |
| GEOLDM (Xu et al., 2023) | 84.4 | **99.3** |
| END (Cornet et al., 2024) | 87.0 | 92.9 |
| UniGEM (Feng et al., 2024) | 85.1 | 98.4 |
| GeoBFN (Song et al., 2024) | 85.6 | 92.08 |
| **EDM-RLPF (ours)** | **87.52** $\pm$ 0.001 | 99.20 $\pm$ 0.06 |
| Data (Ground Truth) | – | 86.5 |

## 5.2 MOLECULE GENERATION ON GEOM-DRUG

**Dataset** Compared to the small molecules in QM9, the GEOM-drug (Axelrod & Gomez-Bombarelli, 2022) dataset consists of more complex molecules with approximately 430,000 conformers. The largest molecule in this dataset contains 181 atoms, with an average of 44.4 atoms per molecule. This makes GEOM-drug a more challenging benchmark for evaluating 3D molecular generation.

**Experimental setup** For this experiment, we fine-tuned the model using the publicly available pre-trained weights of EDM (Hoogeboom et al., 2022). During sampling, we collected 1,024 molecules in total, sampled in batches of 64. Given the larger size and higher complexity of the molecules in the GEOM-drug, we used GFN2-xTB to calculate molecular forces, providing an efficient approximation of the potential energy surface for larger molecular structures. We retained atomic stability and validity as primary evaluation metrics, consistent with previous reports (e.g., EDM and EDN).

**Results** The performance of our method on the GEOM-drug dataset is summarized in Table 2. Compared to the base EDM model, EDM-RLPF improves atom stability from 81.3% to 87.53% and raises validity from 91.9% to 99.20%. While EDM-RLPF achieves the highest atom stability overall,

its validity is slightly lower than GEOLDM, which leads the category. EDM-RLPF improves both molecule stability and validity, suggesting enhanced generation quality for larger molecules.

### 5.3 GENERALIZATION TO OTHER BACKBONES

To evaluate the generalizability of our reinforcement learning framework, we apply **RLPF** to two state-of-the-art generative backbones beyond EDM: **GeoLDM** (Xu et al., 2023) and **UniGEM** (Feng et al., 2024). GeoLDM is a latent diffusion model designed for 3D molecular geometry generation. It introduces an encoder–decoder architecture where a point-structured latent space is constructed to preserve critical roto-translational equivariance properties. Diffusion is performed in this latent space using both invariant scalar and equivariant tensor features. Compared to coordinate-space diffusion, this formulation improves controllability and generation efficiency. UniGEM, on the other hand, unifies molecular generation and property prediction in a diffusion-based framework, using a two-phase generative process to balance both tasks effectively. We conduct molecule generation experiments on the **QM9** dataset using both GeoLDM and UniGEM. Following the same procedure as with EDM-RLPF, we use GFN2-xTB force deviation as the reward signal for fine-tuning. To maintain the integrity of pretrained backbones, we freeze non-diffusion modules (e.g., decoders) during RLPF fine-tuning and update only the diffusion-related parameters.

As shown in Table 3, RLPF consistently improves atom stability, molecule stability, and validity across all backbones. These results demonstrate that RLPF is a versatile reinforcement learning framework that can be flexibly integrated into diverse diffusion architectures, enabling physically grounded fine-tuning for higher-quality molecular generation.

Table 3: Evaluation metrics for 3D molecular generation on QM9 using EDM, GeoLDM, and UniGEM backbones. Metrics include atom stability (A), molecule stability (M), validity (V), and Validity×Uniqueness (V×U). Bold indicates best performance; underline indicates second-best.

| Model | A [%] ↑ | M [%] ↑ | V [%] ↑ | $V \times U$ [%] ↑ |
|---|---|---|---|---|
| EDM (Hoogeboom et al., 2022) | 98.70 | 82.00 | 91.90 | 90.70 |
| **EDM-RLPF (ours)** | $99.37 \pm 0.01$ | $\underline{94.25} \pm 0.13$ | $\mathbf{97.02} \pm 0.08$ | $88.59 \pm 0.04$ |
| GeoLDM (Xu et al., 2023) | 98.90 | 89.40 | 93.80 | $\underline{92.70}$ |
| **GeoLDM-RLPF (ours)** | $\mathbf{99.43} \pm 0.02$ | $\mathbf{95.34} \pm 0.15$ | $\underline{96.28} \pm 0.11$ | $90.66 \pm 0.19$ |
| UniGEM (Feng et al., 2024) | $\underline{99.0}$ | 89.8 | 95.0 | 93.2 – |
| **UniGEM-RLPF (ours)** | $99.17 \pm 0.01$ | $91.28 \pm 0.14$ | $95.57 \pm 0.35$ | $\mathbf{97.8} \pm 0.10$ |

**Discussion on stability–diversity trade-off.** While RLPF consistently improves stability and validity across backbones, we also observe a trade-off: the product metric Validity × Uniqueness (V×U) decreases for EDM and GeoLDM, but increases for UniGEM. We attribute this to how RLPF interacts with the backbone architecture. In EDM and GeoLDM, diffusion jointly operates on coordinates and atom types, so physically grounded rewards encourage the model to focus on narrow high-reward regions, boosting stability but reducing structural diversity. By contrast, UniGEM applies diffusion only on coordinates while predicting atom types once with a frozen head; thus RLPF fine-tunes only the coordinate denoiser, preserving atom-type diversity and yielding higher V×U.

## 6 CONCLUSION

We propose **Reinforcement Learning with Physical Feedback (RLPF)** to fine-tune equivariant diffusion models for 3D molecular generation. By formulating the denoising process as a Markov decision process and optimizing force-field-based rewards, RLPF enhances the quality of generated molecules. Furthermore, RLPF is compatible with various generative backbones, demonstrating strong extensibility across different molecular diffusion architectures.

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

## A    ADDITIONAL STATEMENTS

LLMs were employed during the writing of this paper to polish the text and correct grammatical errors. The prompt used was: "Please detect and correct any grammatical errors in the following text, and polish it to enhance its academic expression. <text>".

## B    ETHICS STATEMENT

This work adheres to the ICLR Code of Ethics. In this study, no human subjects or animal experimentation was involved. All datasets used were sourced in compliance with relevant usage guidelines, ensuring no violation of privacy. We have taken care to avoid any biases or discriminatory outcomes in our research process. No personally identifiable information was used, and no experiments were conducted that could raise privacy or security concerns. We are committed to maintaining transparency and integrity throughout the research process.

## C    REPRODUCIBILITY STATEMENT

We have made every effort to ensure that the results presented in this paper are reproducible. All code and datasets have been made publicly available in an anonymous repository to facilitate replication and verification. The experimental setup, including training steps, model configurations, and hardware details, is described in detail in the paper. Our work is reproducible, and the code is openly available at: https://anonymous.4open.science/r/RLPF-55FC/.

## D    EXTENDED EXPERIMENTS AND ANALYSIS

### D.1    CONDITIONAL MOLECULE GENERATION ON QM9

In this section, we investigate whether the RLPF algorithm improves molecular stability during conditional generation. We evaluated three molecular properties on the QM9 dataset: polarizability ($\alpha$), HOMO-LUMO gap, and LUMO. For dataset partitioning, we follow the same strategy as EDM, splitting QM9 into two subsets, $\mathcal{D}_a$ and $\mathcal{D}_b$, each containing 50,000 samples. The EDM model is first trained on $\mathcal{D}_b$ and then fine-tuned using RLPF with the same configuration as described in Section 5.1, where force deviation computed via GFN2-xTB serves as the primary reward signal.

To guide conditional generation, we incorporate an **augmented reward** that balances physical stability with alignment to the target property. Specifically, the reward is defined over molecular coordinates $x$, atom types $h$, and target property $c$ as:

$$r(x, h, c) = -\lambda \cdot \text{RMSD}_{\text{xTB}}(x, h) - \eta \cdot |\omega(x, h) - c|, \tag{14}$$

where $\omega(x, h)$ is a pretrained property predictor that estimates the property value of the generated molecule. The term $\text{RMSD}_{\text{xTB}}(x, h)$ reflects the deviation from equilibrium as computed using GFN2-xTB, and the term $|\omega(x, h) - c|$ encourages alignment with the target context $c$. Hyperparameters $\lambda$ and $\eta$ control the trade-off between force-based stability and property accuracy.

This composite reward encourages the model to generate molecules that are both physically stable and property-aligned, leading to improved conditional generation performance.

**Results** As shown in Table 4, EDM-RLPF achieves the lowest mean absolute error polarizability ($\alpha$) gap prediction. Specifically, it improves $\alpha$ to 2.29 $Bohr^3$, outperforming all baselines, including GeoBFN. For the HOMO-LUMO gap ($\Delta\varepsilon$) and LUMO energy ($\varepsilon_{LUMO}$), EDM-RLPF also yields clear improvements over EDM, reducing the errors from 655 to 622 meV and from 584 to 521 meV, respectively. These results demonstrate that reinforcement learning with physical feedback enhances property controllability on top of EDM.

**Ablation study on reward weighting.**    To investigate the impact of balancing force stability and property accuracy, we conduct an ablation study on the weighting factor $\eta$ in the augmented reward

Table 4: Mean Absolute Error for molecular property prediction. A lower number indicates a better controllable generation result. Results are predicted by a pretrained EGNN classifier $\omega$ on molecular samples extracted from individual methods. Our method (EDM-RLPF) is fine-tuned using force deviation feedback from the GFN2-xTB force field. The results of QM9 and Random can be viewed as lower and upper bounds of MAE on all properties.

| Property | $\alpha$ | $\Delta\varepsilon$ | $\varepsilon_{LUMO}$ |
|---|---|---|---|
| Units | $Bohr^3$ | meV | meV |
| QM9 | 0.10 | 64 | 36 |
| Random | 9.01 | 1470 | 1457 |
| $N_{atoms}$ | 3.86 | 866 | 813 |
| EDM (Hoogeboom et al., 2022) | 2.76 | 655 | 584 |
| GeoLDM (Xu et al., 2023) | 2.37 | 587 | 522 |
| GeoBFN (Song et al., 2024) | 2.34 | **577** | **516** |
| EDM-RLPF | **2.29** $\pm$ 0.03 | 622 $\pm$ 0.8 | 521 $\pm$ 0.35 |

function defined in Eq. (1), keeping $\lambda = 1.0$ fixed. We evaluate conditional generation on the QM9 dataset for the polarizability ($\alpha$) property using the same setup as in Section D.1.

Table 5: Ablation study on reward weighting for conditional generation of polarizability ($\alpha$). Lower MAE (Mean Absolute Error) indicates better property alignment. Each result is averaged over 3 runs using a pretrained EGNN predictor $\omega$.

| $\eta$ | MAE on $\alpha$ (Bohr$^3$) $\downarrow$ |
|---|---|
| 1.0 | 2.31 $\pm$ 0.03 |
| 0.5 | **2.29** $\pm$ **0.02** |
| 0.1 | 2.79 $\pm$ 0.04 |

We observe that an intermediate value ($\eta = 0.5$) yields the best performance, suggesting that moderate emphasis on property alignment helps optimize controllable generation without sacrificing force stability. Too little weight ($\eta = 0.1$) leads to under-conditioning, while overly strong alignment ($\eta = 1.0$) may interfere with physical consistency.

### D.2 FAIRNESS AGAINST CONTINUED TRAINING

To evaluate whether the performance gains achieved by **RLPF** stem from reinforcement learning rather than from continued training or increased data exposure, we conducted a control experiment on the **QM9 dataset under molecular generation**.

Specifically, we generated **51,200** molecules from the pretrained EDM model and retained only those that passed chemical validity checks (e.g., valency and structural correctness). This number matches the total number of samples generated during the RLPF fine-tuning phase (100 epochs × 512 trajectories per epoch). The accepted molecules were then used as additional training data for further supervised fine-tuning of the EDM model. By keeping the data volume consistent, this control setup allows for a fair comparison, isolating the effect of reinforcement learning from that of simple data augmentation.

**Results:** This experiment demonstrates that while continued training with additional valid data improves diversity-related metrics (such as uniqueness and novelty), it does not yield comparable improvements in structural or force-based stability. The RLPF approach, in contrast, directly optimizes for physically meaningful rewards and produces molecules with significantly better equilibrium stability. These findings underscore the value of reward-guided fine-tuning in RLPF over traditional data-driven augmentation in the QM9 setting.

Table 6: Comparison of EDM-RLPF with supervised fine-tuning using rejection-sampled molecules on QM9. Evaluation metrics include atom stability (A), molecule stability (M), validity (V), uniqueness (U), and novelty (N). Bold indicates best performance.

| Model | A [%] ↑ | M [%] ↑ | V [%] ↑ | U [%] ↑ | N [%] ↑ |
|---|---|---|---|---|---|
| EDM (Hoogeboom et al., 2022) | 98.70 | 82.00 | 91.90 | 90.70 | 65.70 |
| EDM-Continue | $98.99 \pm 0.03$ | $89.47 \pm 0.21$ | $93.20 \pm 0.52$ | $\mathbf{99.36} \pm 0.05$ | $\mathbf{81.45} \pm 0.46$ |
| **EDM-RLPF** | $\mathbf{99.08} \pm 0.05$ | $\mathbf{93.37} \pm 0.25$ | $\mathbf{98.22} \pm 0.15$ | $92.87 \pm 0.07$ | $58.57 \pm 0.24$ |

Table 7: Effect of denoising steps on molecule generation performance (QM9). Evaluation metrics include molecule stability (M), atom stability (A), validity (V), validity × uniqueness (V×U), and novelty (N). For EDM, results at 100/250/500 steps use official checkpoints. Best and second-best results per step are marked in **bold** and underlined, respectively.

| Model | Steps | M [%] ↑ | A [%] ↑ | V [%] ↑ | V×U [%] ↑ | N [%] ↑ |
|---|---|---|---|---|---|---|
| EDM | 100 | $78.01 \pm 0.27$ | $98.00 \pm 0.04$ | $90.15 \pm 0.17$ | $98.76 \pm 0.12$ | $\mathbf{68.03} \pm 0.26$ |
| EDM | 250 | $80.07 \pm 0.10$ | $98.23 \pm 0.07$ | $90.76 \pm 0.10$ | $\mathbf{98.83} \pm 0.06$ | $66.47 \pm 0.43$ |
| EDM | 500 | $80.78 \pm 0.17$ | $98.26 \pm 0.02$ | $91.84 \pm 0.07$ | $98.73 \pm 0.11$ | $66.67 \pm 0.17$ |
| EDM (Hoogeboom et al., 2022) | 1000 | 82.00 | 98.70 | 91.90 | 90.70 | 65.70 |
| END (Cornet et al., 2024) | 100 | 87.40 | 98.80 | 94.10 | 92.30 | – |
| END (Cornet et al., 2024) | 250 | 88.80 | 98.90 | 94.70 | 92.60 | – |
| END (Cornet et al., 2024) | 500 | 88.80 | 98.90 | 94.80 | 92.80 | – |
| END (Cornet et al., 2024) | 1000 | 89.10 | 98.90 | 94.80 | 92.60 | – |
| EDM-RLPF (ours) | 100 | $91.14 \pm 0.07$ | $98.91 \pm 0.07$ | $97.81 \pm 0.25$ | $92.84 \pm 0.34$ | $60.92 \pm 0.60$ |
| EDM-RLPF (ours) | 250 | $92.86 \pm 0.24$ | $99.05 \pm 0.08$ | $98.20 \pm 0.34$ | $92.62 \pm 0.39$ | $58.75 \pm 0.49$ |
| EDM-RLPF (ours) | 500 | $\underline{93.06} \pm 0.26$ | $\underline{99.01} \pm 0.02$ | $\underline{98.30} \pm 0.15$ | $92.41 \pm 0.09$ | $\underline{58.83} \pm 0.15$ |
| EDM-RLPF (ours) | 1000 | $\mathbf{93.37} \pm 0.25$ | $\mathbf{99.08} \pm 0.05$ | $\mathbf{98.22} \pm 0.15$ | $92.87 \pm 0.07$ | $\underline{58.57} \pm 0.24$ |

## D.3 IMPACT OF SAMPLING STEPS ON GENERATION QUALITY

To investigate how the number of denoising steps influences molecular generation quality, we conducted an ablation study on the **QM9 dataset under the molecule generation setting**. We compare our fine-tuned **EDM-RLPF** model against two baselines: the original **EDM** and **END** (Cornet et al., 2024).

For EDM, we evaluated the model at 100, 250, and 500 denoising steps using the official pretrained checkpoints released by the authors. The 1000-step EDM result, as well as all reported END results across different step counts, are extracted directly from their original publications to ensure consistency. For EDM-RLPF, we perform fine-tuning and evaluation using our implementation under the same denoising configurations.

All models are evaluated using five key metrics: molecule stability, atom stability, chemical validity (as computed by RDKit), uniqueness (percentage of unique valid molecules), and novelty (percentage of valid molecules not seen during training). Results are reported as averages over three independent runs where applicable.

Table 7 summarizes the performance comparison. We observe that increasing the number of sampling steps generally leads to improved molecular stability and validity across all models. Notably, **EDM-RLPF consistently achieves the highest molecule and atom stability at every step size**, while maintaining competitive uniqueness and novelty, demonstrating its effectiveness in improving physical plausibility under varying sampling regimes.

## D.4 REJECTION SAMPLING EFFICIENCY WITH RLPF FINE-TUNING

To assess the efficiency gains brought by RLPF, we conducted a rejection sampling experiment under the **QM9 molecule generation setting**. Specifically, we compare the original EDM model with the

EDM model fine-tuned using RLPF. Both models use rejection sampling at inference time, allowing us to isolate the impact of RLPF fine-tuning on sample efficiency.

**Experiment setup:**

- **Goal:** Generate 10,000 stable molecules from either the original EDM or the RLPF-finetuned EDM model.
- **Stability Criterion:** A molecule is considered stable if the RMSD of its atomic forces (computed via GFN2-xTB) is less than 0.2 eV/Å.
- **Sampling Method:** Rejection sampling is applied to filter out unstable molecules. We measure how many total molecules need to be generated—and how long it takes—to collect 10,000 stable ones.

Table 8: Rejection sampling efficiency under molecule generation on QM9. RLPF significantly reduces the number of samples and inference time needed to collect 10,000 stable molecules. Results are averaged over three runs with different random seeds.

| Model | Time (s) ↓ | Molecules Sampled ↓ |
|---|---|---|
| EDM (w/o RLPF) | $1418.45 \pm 24.41$ | $36,400 \pm 346.41$ |
| EDM-RLPF (fine-tuned) | $\mathbf{791.91 \pm 8.22}$ | $\mathbf{19,400 \pm 163.30}$ |

**Results** Despite using the same rejection sampling strategy during inference, the RLPF-finetuned model yields a much higher proportion of stable molecules. This leads to a 44% reduction in sampling time and nearly half the number of samples required, demonstrating the effectiveness of RLPF in enhancing generation efficiency.

### D.5 ABLATION STUDY ON REWARD FUNCTIONS

To better understand how the choice of reward function influences the performance of RLPF, we conducted an ablation study across three reward designs:

- **Stability Reward**: Based on atom valency correctness. See detailed definition below.
- **Force Deviation (xTB)**: Calculated using GFN2-xTB.
- **Force Deviation (DFT)**: Calculated using B3LYP/6-31G(2df,p) DFT.

**Definition of stability reward.** Following the metric proposed by Garcia Satorras et al., we first predict the bond type between each pair of atoms $(i, j)$ based on their Euclidean distance and atomic types. These predicted bonds are used to compute the valency of each atom. A molecule is considered *stable* if every atom satisfies its standard valency constraint.

Let $v_i$ be the predicted valency of atom $i$, and $v_i^{\text{target}}$ be its expected valency based on the atom type (e.g., $v_{\text{C}}^{\text{target}} = 4$ for carbon). A molecule is considered stable if *every atom* in it satisfies the corresponding valency constraint. Formally, we define the molecule-level binary reward as:

$$r_{\text{stable}} = \begin{cases} 1 & \text{if } \forall i \in \{1, \ldots, N\}, \ v_i = v_i^{\text{target}}, \\ 0 & \text{otherwise}, \end{cases} \tag{15}$$

where $N$ is the number of atoms in the molecule.

This binary reward is applied at the final time step of the denoising process and encourages generation of chemically plausible molecules.

We evaluated the fine-tuned models using four metrics: molecule stability, validity, and the product of validity and uniqueness (V×U). Results are reported in Table 9.

Table 9: Impact of reward functions on 3D molecular generation quality on QM9. Metrics include molecule stability (M), atom stability (A), validity (V), validity × uniqueness (V×U), and novelty (N). Bold indicates best performance; underline indicates second-best.

| Reward Type | M [%] ↑ | A [%] ↑ | V (%) ↑ | V × U (%) ↑ | N (%) ↑ |
|---|---|---|---|---|---|
| Stability | **96.45** ± 0.03 | **99.60** ± 0.04 | **98.97** ± 0.07 | 87.74 ± 0.10 | 57.70 ± 0.29 |
| Force (xTB) | **96.45** ± 0.02 | 99.37 ± 0.01 | 97.02 ± 0.08 | 88.59 ± 0.04 | 53.63 ± 0.30 |
| Force (DFT) | 93.37 ± 0.25 | 99.08 ± 0.05 | 98.22 ± 0.15 | **92.87** ± 0.07 | **58.57** ± 0.24 |

**Results** The valency-based stability reward produces high chemical validity and diversity but is less effective at promoting physically meaningful structures. In contrast, rewards based on force deviation—especially those computed via DFT—better align the model with physically plausible configurations while maintaining strong chemical validity. Notably, xTB-based rewards offer similar benefits at significantly lower computational cost, serving as a practical surrogate for DFT. These findings highlight a trade-off in reward design between chemical correctness and physical grounding, and underscore the flexibility of RLPF in supporting diverse objectives.

**Training curves under different reward functions** To further analyze the effect of each reward design on model behavior, we visualize the training dynamics of RLPF with DFT-based force deviation, valency-based stability, and xTB-based force deviation rewards. The plot in Figure 2 shows the evolution of five key metrics—molecule stability, atom stability, validity, uniqueness, and novelty—across training epochs for each reward type.

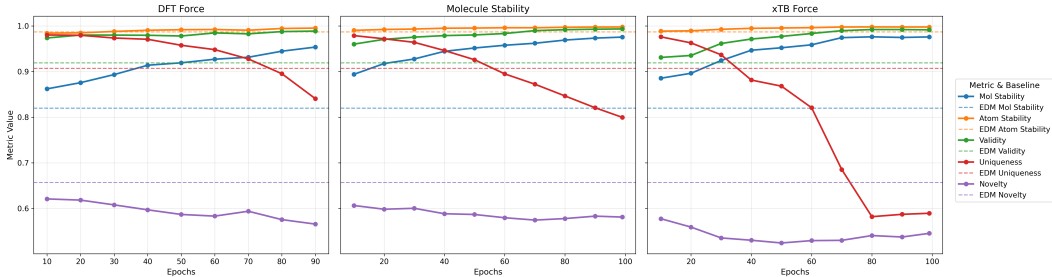

Figure 2: Training curves of generation metrics under three reward types: DFT-based force (left), valency-based stability (middle), and xTB-based force (right). Dashed lines indicate baseline EDM scores for reference.

**Observation** All reward functions lead to increasing molecule and atom stability over time, confirming that RLPF can effectively optimize for structural correctness. Among them, the DFT-based reward achieves strong stability improvements while inducing a relatively smaller drop in uniqueness and novelty. This indicates that DFT feedback better preserves generative diversity while enforcing physical plausibility.

D.6 EFFECT OF CLIPPING THRESHOLD IN RLPF FINE-TUNING

To understand how the PPO clipping threshold $\epsilon$ affects the fine-tuning process in RLPF, we conducted an ablation study under the molecule generation setting on the QM9 dataset. We compared three different values of $\epsilon$: 0.05, 0.2, and 100 (no clipping). All experiments followed the same training setup as described in Appendix E.1.1.

**Observations:**

- When $\epsilon$ is very large (e.g., 100), the reward increases rapidly at first but becomes unstable and collapses in the later stages, as the policy diverges from the pretrained model (shown by the spike in KL divergence).
- Smaller $\epsilon$ values (0.05 and 0.2) result in smoother and more stable training, with consistent gains in molecule stability.

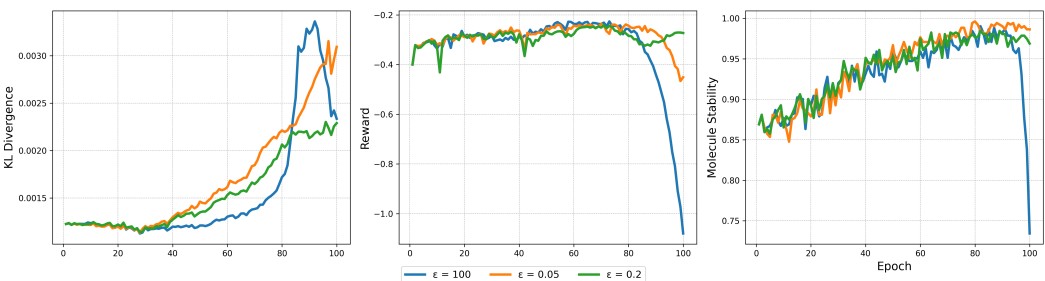

Figure 3: Effect of PPO clipping threshold $\epsilon$ during RLPF training on QM9. Left: KL divergence to pretrained model; Middle: training reward; Right: molecule stability.

- $\epsilon = 0.2$ strikes a better balance between reward improvement and policy stability, making it the default value in our main experiments.

This ablation highlights the importance of policy regularization via clipping in ensuring stable and effective RLPF training.

### D.7 RESIDUAL FORCE ANALYSIS

To further examine the physical plausibility of generated molecules, we evaluated the **residual atomic forces** of sampled conformations. We generated **10,000 molecules** from each model. For the baseline EDM, we used the official pretrained checkpoint; for EDM-RLPF, we evaluated intermediate checkpoints during reinforcement learning fine-tuning (as in Table 1), where the reward was based on DFT force deviations.

Residual forces for each molecule were computed using **DFT** with the **B3LYP functional** and the **6-31G(2df,p) basis set**. We report the **root-mean-square deviation (RMSD)** of atomic forces in **eV/Å**, where lower values indicate geometries closer to equilibrium.

Table 10: Residual atomic forces (RMSD in eV/Å) of generated molecules. EDM-RLPF significantly reduces residual forces compared to the EDM baseline.

| Model | Force RMSD (eV/Å) ↓ |
|---|---|
| EDM | 0.8932 |
| EDM-RLPF | **0.5845** |

**Discussion.** Since our task is *3D molecular generation* rather than explicit conformation optimization, the generated geometries are not guaranteed to be fully relaxed under a quantum force field, and residual forces are not minimized to zero as in geometry optimization. Nevertheless, residual forces serve as a meaningful proxy for physical plausibility. The results above show that EDM-RLPF generates molecules with **significantly lower residual forces** than the EDM baseline, confirming that reinforcement learning with physically grounded rewards encourages the model to produce geometries closer to equilibrium.

### D.8 GEOMETRY OPTIMIZATION ANALYSIS

To further evaluate whether RLPF improves the physical plausibility of generated structures, we compared conformations sampled from the original EDM and our fine-tuned EDM-RLPF (trained with DFT force rewards). For each model, we sampled **1,000 molecules** and optimized them using **DFT with the B3LYP functional and 6-31G(2df,p) basis set**, under a convergence criterion of **fmax = 0.05 eV/Å**. We recorded the following metrics:

- RMSD between pre- and post-optimization structures (lower is better);
- Average number of optimization steps (fewer is better);

- Optimization success rate, i.e., the proportion of molecules converging under the specified criterion.

Table 11: Comparison of DFT geometry optimization efficiency. EDM-RLPF produces molecules that are closer to equilibrium, requiring fewer optimization steps and converging more reliably.

| Model | RMSD (Å) ↓ | Optimization Steps ↓ | Success Rate (%) ↑ |
|---|---|---|---|
| EDM | 0.0981 | 29.14 | 83.6 |
| EDM-RLPF | **0.0482** | **18.58** | **94.4** |

Optimization failures are primarily due to charge imbalance or chemically invalid species (e.g., molecules with inconsistent valence or lacking charge neutrality), which prevent stable SCF convergence. Interestingly, the observed failure rate closely matches the **molecule stability** metric in our main evaluation, confirming that stability is predictive of downstream simulation reliability.

In summary, RLPF not only improves molecular stability but also **reduces reliance on expensive geometry optimization**, leading to more efficient downstream simulations.

### D.9   ADDITIONAL DOCKING EXPERIMENT

To further examine whether the improved stability from our method translates to downstream tasks, we conducted an additional molecular docking experiment. In standard docking workflows, ligands are usually pre-optimized to their lowest-energy conformations before docking (Guedes et al., 2014; Sulimov et al., 2017; Brylinski & Skolnick, 2008). Here, we instead evaluate molecules generated directly by the models, without conformer optimization.

**Setup.**

- **Protein target:** TYK2 (PDB ID: 8S9A).
- **Ligands:** 1,000 molecules generated by (i) the baseline EDM model and (ii) our EDM fine-tuned with RLPF. Both models were trained on QM9.
- **Docking protocol:** Protein prepared with standard preprocessing; ligands docked directly without additional geometry optimization.
- **Metrics:** Average docking score (more negative is better) and docking success rate.

Table 12: Docking performance on TYK2 (PDB: 8S9A). RLPF improves both docking score and success rate.

| Model | Avg. Docking Score ↓ | Success Rate (%) ↑ |
|---|---|---|
| EDM | −4.8363 | 95.4 |
| EDM-RLPF | −4.9438 | **97.7** |

**Discussion.**   These results show that molecules generated by EDM-RLPF yield both a better average docking score and a higher docking success rate compared to EDM. We believe this improvement arises because RLPF encourages generation of molecules that are closer to low-energy, physically stable conformations. Such stability reduces docking errors, accelerates post-processing, and improves the reliability of downstream predictions.

## E   IMPLEMENTATION AND EXPERIMENTAL DETAILS

### E.1   PRETRAINING CONFIGURATIONS

#### E.1.1   THE EDM PRETRAINED ON THE QM9

The EDM model consists of 9 layers, with each hidden layer having a dimension of 256. The SiLU activation function is used, and the Adam optimizer is employed for the optimization process. We set

the batch size to 64 and configured the learning rate to $1 \times 10^{-4}$. For the final model, we selected the version obtained at the 2161st epoch, as this is when the loss reached its lowest value, signaling the end of the pretraining phase.

### E.1.2    THE EDM PRETRAINED ON THE GEOM-DRUG

In this experiment, we directly used the GEOM-drug model parameters provided by EDM. On the GEOM dataset, EDM is trained using EGNNs with 256 hidden features and a 4-layer architecture. The models were trained for 13 epochs, which corresponds to approximately 1.2 million iterations with a batch size of 64. The pretrained weights are available at: https://github.com/ehoogeboom/e3_diffusion_for_molecules/tree/main/outputs/edm_geom_drugs.

### E.2    FINE-TUNING WITH RLPF

We fine-tuned the pretrained EDM models on both the QM9 and GEOM-drug datasets using our proposed RLPF framework. All experiments were conducted under the molecule generation setting. The reward signals were derived from either force deviations (computed using DFT or xTB) or valency-based stability checks.

**Training setup**    For QM9, the fine-tuning was performed over 100 epochs, sampling 512 molecules per epoch, yielding a total of 51,200 molecules. The model was optimized using the AdamW optimizer with a learning rate of $1 \times 10^{-5}$. The hyperparameters were set to $\beta_1 = 0.9$, $\beta_2 = 0.999$, $\epsilon = 1 \times 10^{-8}$, and a weight decay of $1 \times 10^{-4}$. To ensure stable policy updates during reinforcement learning, we used PPO with a clipping threshold of $\epsilon = 0.2$ (see Eq. equation 11) and applied advantage normalization with a clipping range of 1.0.

**Handling invalid structures.**    When using DFT-based force deviation as the reward, certain invalid structures occasionally caused failures in the force calculation (e.g., due to unbalanced charge or highly distorted geometry). In such cases, we assigned a fixed penalty reward of $-5$ to reduce their impact on training.

**Parallel sampling and reward computation.**    As shown in Figure 4, reward evaluation does not require GPU acceleration. To maximize throughput, we employed pipeline parallelism between sampling and reward computation. Molecule batches were generated on GPU and immediately dispatched to CPU-based workers for reward calculation, allowing both stages to proceed concurrently.

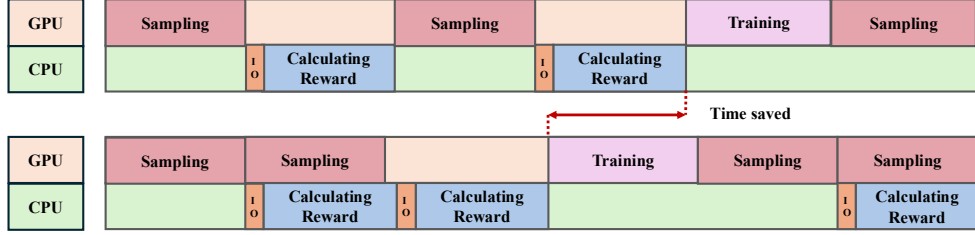

Figure 4: Schematic diagram of pipeline parallelism between sampling and reward evaluation.

**GEOM-drug fine-tuning.**    For the larger GEOM-drug dataset, we fine-tuned EDM using rewards computed from GFN2-xTB. Due to the high computational cost of DFT on large molecules, DFT-based rewards were not used in this setting. Each round of fine-tuning involved sampling 1024 molecules, repeated over 2 rounds. The use of xTB strikes a balance between computational efficiency and physical accuracy, making it suitable for more complex molecules.

**Diversity and stability trade-off.** To evaluate diversity, we report novelty following EDM's standard protocol. On QM9, RLPF slightly reduces novelty compared to the pretrained EDM (58.6% vs. 65.7%), suggesting that the model becomes more concentrated around the training manifold, thereby improving stability (see Table 1) at the cost of reduced exploration.

Notably, as observed by Vignac & Frossard (2022), the QM9 dataset constitutes a near-complete enumeration of stable molecules under certain constraints. In this context, excessively high novelty may reflect divergence from the true data distribution, increasing the likelihood of generating chemically implausible structures. Therefore, a moderate drop in novelty may actually indicate better alignment with valid chemical space, rather than a loss of model quality.

Table 13: Diversity evaluation on QM9. We report novelty among valid and unique molecules. Results are averaged over three runs.

| Model | Novelty (%) ↑ |
|---|---|
| EDM (Hoogeboom et al., 2022) | 65.7 |
| GeoLDM (Xu et al., 2023) | 57.0 |
| GeoBFN (Song et al., 2024) | **66.4** |
| EDM-RLPF (ours) | $58.57 \pm 0.25$ |

**Training convergence criterion** We define convergence in RLPF fine-tuning based on either:

- Reaching a maximum of 100 epochs, or
- The average reward entering a stable, high-performance range.

These convergence thresholds vary depending on the reward type:

- For valency-based stability rewards, training is considered converged when the reward exceeds **0.95**.
- For force-based rewards (e.g., xTB RMSD), convergence is reached when the reward exceeds **-0.25**.

Figure 5 shows representative reward curves under both reward types. These thresholds were determined empirically by observing when the reward plateaus or when continued training yields diminishing returns.

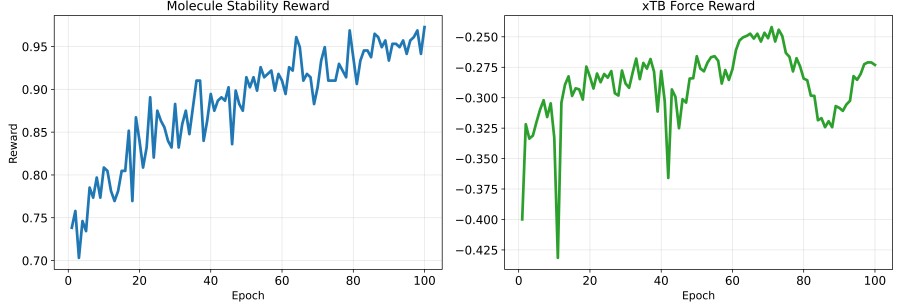

Figure 5: Comparison of training reward curves under valency-based stability (left) and xTB-based force (right) reward functions.

### E.3 COMPUTATION COST ANALYSIS

We report the computational resources required for the fine-tuning of RLPF in different settings.

**QM9 (RLPF with DFT and xTB).** When fine-tuning the EDM model in the QM9 dataset using RLPF with DFT-based force deviation rewards, we utilized 13 NVIDIA H100 GPUs for model

training and 500 CPU cores in parallel for reward computation with the B3LYP/6-31G(2df,p) method. In contrast, when using GFN2-xTB for force evaluation, the reward computation required only 15 CPU cores in parallel, significantly reducing the computational burden while still providing effective training signals.

**GEOM-drug (RLPF with xTB).** For the larger and more complex GEOM-drug dataset, we fine-tuned the EDM model using RLPF with GFN2-xTB-based rewards. The training process required 10 NVIDIA H100 GPUs, while the reward computation was performed efficiently using 15 CPU cores. DFT-based rewards were not used in this setting due to their prohibitively high computational cost on large molecules.

These results highlight the scalability of RLPF: while high-fidelity rewards (e.g., DFT) are computationally expensive, approximate methods like xTB offer a practical trade-off between accuracy and efficiency, particularly in large-scale settings.

## F    FULL WORKFLOW OF RLPF

For completeness, we provide the full workflow of the RLPF algorithm that outlines its procedural structure and training loop. While the high-level logic is illustrated in Figure 1 and Algorithm 1 in the main text, the following summary describes each step in detail:

1. **Sample Trajectories**: The pre-trained model $p_{\theta_{\text{old}}}$ is used to generate $K$ molecular trajectories by denoising latent variables over $T$ timesteps. This captures both the intermediate states $z_t$ and the final molecular structure $(x, h)$.

2. **Calculate Rewards**: The generated molecules $(x, h)$ are evaluated using physically grounded reward functions, such as DFT- or xTB-based force deviation, or valency-based stability. These values serve as scalar rewards $r(x, h)$.

3. **Fine-tune with RL**: For each trajectory $k$, the reward $r(x^k, h^k)$ is normalized to obtain an advantage estimate $\hat{A}_t^k$. The importance sampling ratio $I_t^k(\theta)$ is computed using log-likelihood scores from Section 4.5. A PPO-style clipped policy objective is optimized to update $\theta$.

This pipeline is repeated across multiple epochs in an online fashion, alternating between generation and policy improvement.

## G    LIMITATION OF RLPF

While RLPF significantly improves molecular stability through reward-based fine-tuning, its effectiveness hinges on a crucial assumption: the base generative model must be capable of producing a wide range of samples, including both high-quality and low-quality molecules. This diversity is essential for the advantage estimation step in reinforcement learning, where advantages are computed using normalized returns based on mean and variance.

If the pretrained model fails to generate a sufficient number of poor or unstable samples, the estimated advantages across trajectories may become uniformly small. As a result, the gradient updates derived from the reinforcement signal will have limited impact, and RLPF may offer only marginal improvements. In practice, we observe that RLPF is most effective when applied to a base model that exhibits moderate performance—sufficiently stable to ensure chemical validity, yet diverse enough to expose room for reward-guided improvement.

This limitation highlights the importance of sampling diversity in reward-based fine-tuning, and suggests that future work could explore adaptive weighting or trajectory selection strategies to mitigate this sensitivity.

## H SAMPLES FROM FINE-TUNED MODELS WITH RLPF

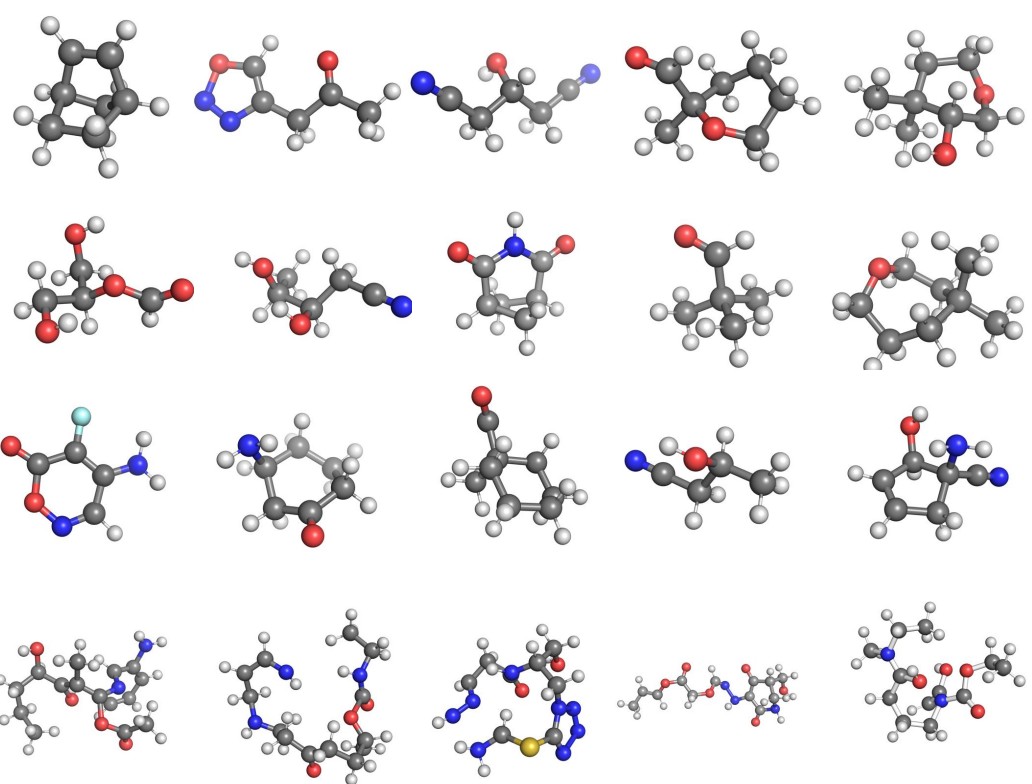

Figure 6: Representative samples generated after RLPF fine-tuning. The top row shows molecules from EDM pretrained on QM9 and fine-tuned using DFT-based force rewards. The second row shows results using valency-based stability as rewards. The third row depicts fine-tuning with GFN2-xTB force-based rewards. The bottom row shows molecules generated from EDM pretrained on GEOM-drug and fine-tuned using xTB forces. RLPF consistently improves structural stability and equilibrium quality across all reward types.

