# OpenReview forum: "Guiding Diffusion Models with Reinforcement Learning for Stable Molecule Generation"
_ICLR.cc/2026/Conference — ICLR 2026 Conference Withdrawn Submission_

### Official Review · Reviewer_ygra · 2025-10-20

**Soundness:** 2
**Presentation:** 2
**Contribution:** 1
**Rating:** 2
**Confidence:** 4

**Summary:**

In this paper, the authors propose to adapt reinforcement learning with physical feedback to fine-tune diffusion models, in order to produce equilibrium structures adhering to physical principles. The method extends DDPO to the molecular domain, employing a physics-informed reward function derived from force fields evaluations, that guides the generation toward meaningful configurations. The authors demonstrate the effectiveness of the approach through various experiments on QM9 and GEOM-drug datasets, and using two different pretrained diffusion models.

**Strengths:**

- This paper is well-written and well-structured.
- The authors incorporate physics-informed reinforcement learning to guide diffusion models to generate physically plausible outputs.
- The authors validate the effectiveness of the proposed approach by showing an improved performance across different datasets, and combined with different pretrained diffusion models.

**Weaknesses:**

- The paper offers limited methodological novelty. It primarily builds on top of prior work [1], that adapts DDPO to optimize pre-trained diffusion models for various downstream objectives. This paper simply applies the same methodology to the molecular domain, using a different objective function defined over generated 3D molecular structures.
- The impact of the masking mechanism introduced in Equation 13 on performance remains unclear. Could the authors provide additional experiments to support the claim that this masking stabilizes training?
- What is the rationale behind using the reward function in Equation 12? Have you explored alternative physics-informed reward functions? And if so, how do they compare?

**Questions:**

- How does the number of sampled trajectories affect the performance?
- Could you report the molecule stability and uniqueness of the generated samples for GEOM-drug (Table 2)?
- How does the proposed RLPF approach affect different properties of the generated samples, such as QED, SA and LogP, as measured in [2]?

[1] Black, Kevin, et al. "Training diffusion models with reinforcement learning." arXiv preprint arXiv:2305.13301 (2023).
[2] Hoogeboom, Emiel, et al. "Equivariant diffusion for molecule generation in 3d." International conference on machine learning. PMLR, 2022.

---

### Official Review · Reviewer_nARo · 2025-10-29

**Soundness:** 3
**Presentation:** 3
**Contribution:** 2
**Rating:** 2
**Confidence:** 3

**Summary:**

The paper presents a policy optimization RL-based method for adaptation of a pre-trained diffusion model to increase the stability of generated molecules by employing physics-based reward feedback.

**Strengths:**

- I am not aware of works specifically focused on fine-tuning diffusion models for improving molecular stability (although I am not an expert in this domain), which to my knowledge  is an important task from a chemical standpoint.
- The paper is easy to follow and sufficiently formal to be clear.
- Experimental evaluations encompass a wide range of pre-trained models as baselines.

**Weaknesses:**

- (main concern) although the paper shows an interesting application, I believe that there isn't nearly any core methodological innovation. Concretely, I would not regard RLPF as a 'novel framework' as stated within the abstract and along the paper, but rather just a specific choice of reward function and lower-level engineering/implementation details.  The algorithmic machinery employed is already very well established in this field after it has been introduced ~3 years ago with hundreds of works extending it, and this paper seems to present a quite straightforward application.
- I do not agree with the way the paper is written in multiple parts, as it seems to present RLPF as a novel algorithmic framework rather than a specific choice of reward function for a very established class of methods. Portraying this fact properly, and presenting the paper as an applied contribution rather than methodological, which I believe to be essential, would require a significant rewriting of the manuscript.

In short,  the proposed solution seems to use very established methods and not providing sufficient novel methodological or conceptual contributions for the standards of this conference. Similarly, it seems to me obvious that this applied problem can be solved in such way, as a consequence I do not see sufficient novelty even from an applied/practical standpoint regarding how to tackle a practically-relevant task.

I currently believe this work would be a good fit for a specific ML workshop, or a chemical journal if experimental evaluation is significantly expanded (e.g. beyond standard datasets such as QM9 or GEOM-Drugs). Relevance for an ML conference might instead arise, for instance, by comparing the performances of multiple established fine-tuning schemes (including the one currently used), or proposing novel algorithmic ways to leverage the physical feedback extending non-trivially existing schemes.

**Questions:**

Did I misinterpret/misunderstood any fundamental aspect mentioned within my weakness points?

---

### Official Review · Reviewer_n1wD · 2025-10-31

**Soundness:** 3
**Presentation:** 3
**Contribution:** 2
**Rating:** 2
**Confidence:** 3

**Summary:**

This paper proposes post-training an equivariant diffusion model using reinforcement learning with physical feedback (RLPF), with the atomic force RMS index calculated by force field/quantization as the reward. Policy updates are performed using DDPO/PPO-style pruning objectives, and improvements in stability and effectiveness are reported on QM9 and GEOM-drug. The paper also demonstrates transferability and a series of ablation techniques across different backbones, including EDM, GeoLDM, and UniGEM.

**Strengths:**

1.An explicit reward function (forced RMS) oriented towards "structural physical feasibility" is used to cast the diffusion trajectory into an MDP and train it stably in a PPO style. This approach is reasonable and highly consistent with the domain requirements for stable conformations.

2.It covers multiple backbones and various ablation methods, including reward type, sampling steps, pruning threshold, etc., and includes a control group for continued training and RL, demonstrating a strong awareness of research issues.

3.It provides anonymous code links and a reproduction appendix, demonstrating a good awareness of reproducibility.

**Weaknesses:**

1.There is a lack of systematic comparisons with energy/force field guided sampling (non-RL) methods (such as energy guidance, score distillation with energy, and consistency comparisons with molecular force field post-processing); the condition generation uses a predictor ω to evaluate target properties, which carries the risk of evaluator coupling or bias.

2.Equation (12) defines the RMS of the force vector (the unit should be eV/Å or kJ/mol/Å). The text repeatedly refers to it as "RMSD of forces" (RMSD is usually used for geometric coordinates), which is inappropriate terminology and lacks consistent unit specification, especially since a threshold of 0.2 eV/Å is given in D.4. It is recommended to consistently use "RMS force" and clearly specify the unit and calculation details when it first appears in the manuscript.

3.GEOM-drug was evaluated on only 1024 samples and only reported Atom Stability and Validity, which is asymmetric with the more comprehensive indicators of QM9. Therefore, the evidence for the conclusion that "it also improves large molecules" is insufficient.

4.In Table 1 (EDM-RLPF, QM9, DFT reward), V×U=92.87, while in Table 3 (EDM-RLPF, QM9, seemingly xTB reward), V×U=88.59. The text's discussion of "RLPF leading to a diversity trade-off" is inconsistent with the two tables.

5.The last part of equation (13) states that the Gaussian normalization constant log(2πσ^2) "cancels out" when calculating the ratio of pθ / pθ_old. This is predicated on σ and the constant being independent of θ or being identical. However, in the case where σ_ij is predicted by the network (or noisy scheduling)/depends on θ, this term will not cancel out; ignoring it will change the ratio and thus the PPO objective, which is an important theoretical detail error/unclear. The source of σ (fixed scheduling? learnable head?) and the precise implementation of the ratio need to be clarified.

**Questions:**

1.The author equates "stable structure" with a small RMS (Relaxed Minimum) under stress. Please discuss the differences between this and the minimum/local minimum energy, and the experimentally observable conformation; why was stress chosen instead of energy difference (ΔE to relaxed) or a combination of energy and stress as the primary reward?

2.Compared to non-RL methods with energy/classifier guidance (such as directly guiding the sampling trajectory with energy/force field), what are the advantages, disadvantages, and costs of RLPF? Can an energy-guided baseline be increased within the same budget?

3.Equation (5) The parameterization and source (α, σ) of the sampling update should be strictly aligned with the original EDM text and the notation should be consistent; is the noise of the same prediction coordinate and category channel?

4.Does the importance ratio in DDPO include variance/normalization constant? If σ is learnable, how does the statement "constant cancellation" hold true in the text?

5.Why not use GAE for advantage calculation? With only terminal reward settings, does having the same advantage at each step lead to large gradient variance/slow convergence? Should we try baseline value function or KL reward-penalty stable training?

6.For QM9-DFT (CPU-only), please provide a detailed computing power budget (number of cores/nodes/total duration/failure rate) and robustness (different DFT settings, different SCF thresholds).

---

### Official Review · Reviewer_JsJx · 2025-11-01

**Soundness:** 3
**Presentation:** 3
**Contribution:** 1
**Rating:** 2
**Confidence:** 3

**Summary:**

This paper proposes Reinforcement Learning with Physical Feedback (RLPF), a framework to fine-tune 3D equivariant diffusion models for molecular generation. RLPF adapts the Denoising Diffusion Policy Optimization (DDPO) algorithm, formulating the denoising process as an MDP. The key contribution is the use of reward functions derived from physical force fields (e.g., DFT or GFN2-xTB) to guide the model toward generating more energetically stable and physically plausible structures. Experiments on the QM9 and GEOM-drug datasets demonstrate that RLPF significantly improves molecule stability, chemical validity, and the efficiency of downstream tasks (like rejection sampling and geometry optimization) compared to the baseline diffusion models.

**Strengths:**

The paper addresses a critical and practical problem in 3D molecular generation: the physical instability of generated structures. The demonstrated improvements in downstream efficiency (e.g., a 44% time reduction in rejection sampling (Table 8) and 36% fewer optimization steps (Table 11)) are significant and highly valuable for practitioners.

The work successfully demonstrates that adapting an existing RL framework (DDPO) with a domain-specific, physics-based reward is a viable and effective strategy for "polishing" the output of 3D diffusion models to align with desired physical properties.

The RLPF framework appears to be model-agnostic, showing consistent improvements when applied to multiple different diffusion backbones (EDM, GeoLDM, and UniGEM), as shown in Table 3.

The paper is generally well-written and clearly structured. The methodology, building upon the established DDPO framework, is straightforward to follow.

**Weaknesses:**

The core RL algorithm is a direct application of the existing DDPO framework. The proposed "Size-Invariant Log-Likelihood Estimation" (Sec 4.5) appears to be a standard masking technique used in sequence processing. Furthermore, using physical properties as reward signals is a known concept in other areas of molecular generation (e.g., with autoregressive models). The primary contribution is the combination of these existing ideas, rather than a novel RL algorithm or reward formulation.

The paper's most significant weakness is that the gains in stability come at a considerable cost to generative diversity. This is evident in the decreased Validity × Uniqueness (V×U) and Novelty scores across most experiments (e.g., Table 3, 6, 9, 13). For de novo generation, where discovering novel and diverse structures is paramount, this is a major limitation. The work would be stronger if it addressed or attempted to mitigate this trade-off more directly.

The primary comparison is against the baseline models (e.g., EDM). However, fine-tuning a model with a stability reward is almost guaranteed to improve that specific stability metric. A more compelling benchmark would involve comparing RLPF to other methods designed to enhance stability, such as other RL-based approaches (if any) or simpler fine-tuning strategies (e.g., filtering the dataset for stable molecules, as in Sec D.2, but perhaps with a stricter filter).

The reward function ablation (Table 9) raises questions. The most computationally expensive reward (DFT) yields *worse* molecule stability (93.37%) than the much cheaper xTB (96.45%) and Stability (96.45%) rewards. While the text notes DFT preserves diversity better (V×U=92.87%), this makes the main result in Table 1 (which uses DFT) seem less impressive, as a simpler reward could have achieved higher stability. This suggests the high cost of DFT is primarily buying back diversity lost by RL, rather than maximizing stability.

**Questions:**

The drop in diversity (V×U, Novelty) is a primary concern. Have the authors considered methods to explicitly manage this trade-off? For instance, incorporating a KL-divergence penalty against the original pre-trained model (similar to PPO-KL or how DPO works) could prevent the policy from deviating too far and collapsing to a narrow, high-reward mode.

Could the authors elaborate on the findings in Table 9? It is counter-intuitive that the cheaper xTB and Stability rewards achieve higher molecule stability (M=96.45%) than the high-fidelity DFT reward (M=93.37%). Why does the supposedly more accurate DFT reward lead to lower stability? Does this imply the DFT reward signal is noisier, or is it optimizing for a different objective that is less correlated with the "Molecule Stability" metric?

Section D.2 compares against "EDM-Continue." Could a stronger baseline be a model fine-tuned *only* on the subset of the original data that passes a strict stability check (e.g., force RMSD < 0.2)? This would be a non-RL method for achieving the same goal and would provide a valuable point of comparison for the added complexity of RLPF.

I am willing to raise my score if all concerns are addressed.

---

### Note · Authors · 2026-01-20

I have read and agree with the venue's withdrawal policy on behalf of myself and my co-authors.